# The Corrosion Resistance and Mechanism of AT13/Fe-Based Amorphous Composite Coatings

**DOI:** 10.3390/mi13010056

**Published:** 2021-12-29

**Authors:** Zhenhua Chu, Shikun Teng, Yuyun Zhou, Xingwei Zheng, Jingxiang Xu, Fang Wang, Baosen Zhang

**Affiliations:** 1Department of Mechanical Engineering, College of Engineering, Shanghai Ocean University, Shanghai 201306, China; shikunteng@163.com (S.T.); yyunzhou2021@163.com (Y.Z.); xjxu@163.com (J.X.); wang_fangshou@163.com (F.W.); 2College of Science, Donghua University, Shanghai 201620, China; 3School of Materials Science and Technology, Nanjing Institute of Technology, Nanjing 211167, China; pantherf@163.com

**Keywords:** Fe-based amorphous coating, AT13, plasma spraying, corrosion resistance

## Abstract

Due to high strength, high wear resistance and high corrosion resistance, the amorphous metallic glasses were investigated widely. In the present study, the corrosion resistance of amorphous coating and composite coatings with various proportions of AT13 (Al_2_O_3_–13 wt.% TiO_2_) ceramic as additions in 3.5 wt.% NaCl solution were studied. The corrosion resistance was improved obviously as the addition of AT13, and when the content of AT13 was 15 wt.%, the composite coating had the lowest corrosion current density (1.75 × 10^−6^ A cm^−2^) and the highest corrosion potential (−411 mV), which was 5.14 × 10^−5^ A cm^−2^ and −580 mV for Fe-based metallic glassy coating, respectively. The corrosion mechanism was proposed according to the long-time immersion corrosion test.

## 1. Introduction

Although bulk metallic glasses (BMGs) exhibit ultrahigh hardness and strength, and superior corrosion resistance and wear resistance [1,2,3,4], metallic glasses are seldom used in engineering because of the limitation of the glassy formation ability. Fortunately, the metallic glassy coating can be fabricated by the spraying method [5]. It is required to put micron powders together, and then it can enlarge its size at three-dimensions. However, due to the deformation mechanism of bulk metallic glasses, which depends on the development of shear bands, it exhibits poor plastic deformation. As a result, a great amount of research efforts have been devoted to fabricating bulk metallic glassy composites (BMGCs) to improve the plasticity of BMGs by introducing ductile metallic crystalline or ceramic phases with high strength into the metallic glassy matrix in order to form multiple shear bands.

It is reported [6] that the amorphous composite coatings can improve the bonding and impact resistance of the coatings. Wang et al. [7] found that the tungsten carbide (WC) phase-reinforced Fe-based amorphous coating is unstable in alkaline environment and then chemically reacts to dissolve, diminishing the corrosion resistance. Cai et al. [8] improved the wear resistance of the coating via enhancing the bonding strength between the ceramic particles and the matrix. In our previous studies [9,10], the excellent wear resistance and corrosion resistance were obtained in composite coatings. Electrochemical corrosion studies were performed to further investigate the effect of the AT13 (Al_2_O_3_–13 wt.% TiO_2_) ceramic phase on the fabricated materials in different aggressive media and the long-term corrosion mechanism. In the previous study [11], ZrO_2_ ceramic powders were added into the Fe-based metallic glassy coating by a gas multiple-tunnel, due to the high melting temperature of ZrO_2_. In this study, AT13 ceramic was adopted, due to its lower melting temperature, to fabricate a composite Fe-based metallic glassy composite coating by mixing Fe-based metallic powders and AT13 ceramic powders as feedback powders. It is an easy method to obtain composite coatings, and it is worth noting that the ability of formation of amorphous is not affected by the addition of AT13. 

In this paper, the coating corrosion resistance in different media and the effect of the second-phase content on the electrochemical corrosion performance of the coatings were studied. Moreover, the long-term corrosion behavior of the amorphous composite coating was evaluated in NaCl solution. It was observed that when the content of AT13 was 15 wt.%, the coating composite had the lowest corrosion current density and the highest corrosion potential.

## 2. Experimental Procedure

Fe54Cr25Mo17C2B2 (atomic ratio) amorphous powders were produced by high-pressure atomized Ar gas. Then, different weight fractions (5–20%) of Al_2_O_3_–13%TiO_2_ (AT13) ceramic powders (20–40 μm) were mixed with amorphous powders in a mechanical mixing machine for 4 h. The mild steel (0.45 wt.% C) was selected as a substrate, with a size of 10 mm × 10 mm × 12 mm. All substrates were machined and polished, then degreased by acetone, dried in air and grit-blasted prior to the deposition. The coating was fabricated by the plasma-spraying process. 

The microstructures of powders and the as-sprayed coating were examined by scanning electron microscopy (SEM, Hitachi S-4800, Japan) coupled with energy dispersive spectroscopy (EDS) and X-ray diffraction (XRD, Bruker D8 Focus). In the present study, we depended on the SEM photos and software of Image-Plus to obtain the porosity statistical results. Here, more than 6 images were chosen to count the porosity.

Electrochemical measurements were performed at room temperature in a three-electrode cell, utilizing the saturated calomel electrode (SCE) and the graphite electrode as the reference and auxiliary electrodes, respectively. Specimens for the corrosion test were closely sealed with epoxy resin, leaving only an end-surface with a surface area of 1 ∗ 1 cm^2^ exposed for testing. The electrochemical behavior was characterized by recording potentiodynamic polarization at a potential sweep rate of 0.5 mVs^−1^ from −100 to 1500 mV in 3.5 wt.% NaCl solution open to air after immersing the specimens for an hour. Each test was repeated three times for repeatability and reliability. In addition, electrochemical impedance spectroscopy (EIS) was conducted at the OCP in the frequency range from 100 kHz to 0.01 Hz with a sinusoidal amplitude of 10 mV for 28 days. The impedance plots were interpreted on the basis of the equivalent circuit using a suitable fitting procedure by Echem Analyse. 

For composition analysis in passivation film formed after electrochemical treatment, a specimen was potentiodynamically polarized into the middle of the passive region and then taken out immediately for X-ray photoelectron spectroscopy.

## 3. Results

### 3.1. Characterization of Powders and As-Sprayed Coatings

The XRD spectra of amorphous composite coatings are shown in Figure 1. It can be seen that a broad diffraction bump appeared at the angle of 2θ = 43°~45° of all coatings. Moreover, the crystal peak corresponding to the γ-Al_2_O_3_ phase also appeared in the spectrum of the composite coatings, and the intensity of the diffraction peak increased with the increase of the added content.

The cross-section morphologies of the Fe-based amorphous coating with 10 AT13 wt.% are shown in Figure 2. It was found that the as-sprayed coating is closely bonded to the substrate without obvious cracks. The additions are homogenous, as shown in Figure 2b, which is a magnified image of the region marked in Figure 2a, and have a dense structure with a thickness of about 250 μm. In addition, the porosity of the coating is about 1.51%. According to our previous study [11], it is indicated that the silver-grey phase in the coating is alumina phase, and the grey-black flat-shaped phase is mainly Fe-based amorphous phase, as marked in Figure 2b.

### 3.2. Corrosion Resistance Measurements

#### 3.2.1. Open Circuit Potential (OPC) Test

To investigate the corrosion resistance of Fe-based amorphous composite coatings in 3.5 wt.% NaCl solutions, immersion test method was chosen. Figure 3 shows the relationship between the open circuit potential of different samples against time immersed in a 3.5 wt.% NaCl solution for 3600 s. All the curves of five kinds of coatings showed two stages: there is a gradual OPC decrease as time goes on, followed by a slow OPC change thereafter. With the increase of AT13 content, the potential of coatings gradually increased. When the additive content was 20 wt.%, the OPC was slightly lower than 15 wt.% AT13. The reason is related to the porosity of the coating. The corrosive medium penetrated into the coating through the pore and the activation corrosion process continued.

#### 3.2.2. Potentiodynamic Polarization Test

Figure 4 displays the potendiodynamic polarization curves of different samples in 3.5 wt.% NaCl solutions. Table 1 shows some electrochemical parameters, such as corrosion potential (Ecor), corrosion current density (icor), passive current density (ipass), and transpassive potential (Etr). The results indicate high corrosion resistance of Fe-based composite coatings with an obvious activation zone, transition zone, passivation and over-passivation zone. Due to the amorphous structure of the Fe-based coating, the distribution of elements is homogeneous. Meanwhile, the weight ratios of Mo and Cr in the Fe-based amorphous coating are 27.22% and 21.69%, respectively. The Mo element improves the formation of the passive film Cr_2_O_3_. In 3.5 wt.% NaCl solution, the amorphous coating and composite coatings presented excellent corrosion resistance, and the self-corrosion potential was from −580 to −411 mV. Corrosion current density was in the order of 10^−5^~10^−6^ Acm^2^. Moreover, with the addition of the AT13 ceramic phase, the self-corrosion potential of the composite coatings was positively shifted compared with the pure Fe-based coating, and the passive current density was sharply reduced. It shows that the corrosion resistance of the composite coating with AT13 ceramic phase improved. Moreover, it is worth noting that the breakdown potential was over 1 V versus SCE, implying that the oxygen evolution appeared. 

Figure 5 depicts the surface corrosion morphologies of the amorphous coating and the 15 wt.% AT13 composite coating after dynamic potential scanning to the over-passivation interval in NaCl solution. The severe corrosion of the Fe-based amorphous coating in the spray pores and the edges of un-melted particles was clearly seen (Figure 5a). Besides, amounts of evenly distributed corrosion holes appeared on the surface (marked with a white dotted circle). The results demonstrate that the coating mainly produced local pitting corrosion, the passivation film has failed and the coating matrix dissolved rapidly. It can be seen that the corrosion of the composite coating is obvious in holes and crevices, and the number of corrosion holes on the surface of the composite coating was significantly less than that of the pure coating (Figure 5b), indicating that the composite coating has a stronger resistance to local corrosion than the pure amorphous coating. 

## 4. Immersion Test

### 4.1. EIS Measurements

Potentiodynamic polarization and electrochemical impedance spectroscopy (EIS) measurements were performed to evaluate the corrosion resistance performances of coatings. The above experimental results show that the coatings exhibited excellent corrosion resistance in 3.5 wt.% NaCl solution. In order to better evaluate the corrosion failure process of different coatings, long-term immersion experiments in 3.5 wt.% NaCl solution combined with EIS measurements were carried out to analyze the corrosion behavior of the coatings.

Figure 6 illustrates the EIS plots of different coatings completely immersed in 3.5 wt.% NaCl solution for 28 days. The Nyquist plots recorded at different periods of immersion time of the coatings are shown in Figure 6a,c,e,g,i and Figure 6b,d,f,h,j are the corresponding plots of Bode curves. There exist two stages of the corrosion process of coatings: 0 and 1–28 days. The data recorded at the initial stage of corrosion exhibit two time constants due to the relatively high porosity of the coating. At the initial stage of immersion, Cl^−^ ions present in the solution penetrated into the coating from cracks and holes [12]. At the same time, the chrome oxide was formed, and it improved the corrosion resistance of the coating. 

For a single coating, the impedance value was reduced from 1705 to 731 Ω·cm^2^, and the impedance value of the 5 wt.% AT13 composite coating was reduced from 2587 to 2160 Ω·cm^2^. It is difficult to distinguish the two time constants since the capacitance arcs of the low frequency are not clear [13]. One time constant was in the high-frequency range related to the capacitance impedance of the coating. The other constant was in the low-frequency range related to charge transfer resistance and a double-layer capacitance between the solution/coating interface. As the immersion time increased, the value of the resistance changed within a small range. During the 0-day immersion, the Bode plots (Figure 6b,d) presented a maximum. With the immersion increasing, the high-frequency phase angle shifted to the left and the curve dropped, indicating that more and more defects would be formed in the coating due to corrosion. In Figure 6e, the Nyquist impedance plots of the 10 wt.% AT13 composite coating showed two continuous semi-circle shapes. With the immersion increasing, the radius of the capacitance arc increased gradually, suggesting that the porosity of the coating was further reduced and the resistance of the passivation membrane to the electrolyte was enhanced. With the immersion time to day 10, the resistance value decreased gradually and then fluctuated within a small range. The EIS plots of 15 wt.% and 20 wt.% AT13 composite coatings in 3.5 wt.% NaCl solution for the 28 days are shown in Figure 6g–j. There existed two stages of the corrosion progress for the coatings: 0 days and after 1 day. At day 0, a single capacitance resistance arc occurred in Nyquist impedance plots (shown in Figure 6g,i). It can be seen that the complete shielding layer existed in the composite coating, which effectively prevented the corrosion. The resistance value of the 15 wt.% AT13 composite coating was 2780 Ω·cm^2^ at the same time. Compared to the 20 wt.% AT13 composite coating, the radius of the semi-circle arc was bigger, indicating that the corrosion resistance of the 15 wt.% AT13 composite coating was better. After 1 day of immersion, the Bode impedance plots showed two time constants, indicating that the solution had reached the oxide layer interface and the micro-galvanic reaction. From the EIS plots of the pure amorphous coating and the composite coating, we can learn that the radius of the capacitance resistance arc was the biggest when the addition of AT13 was 15 wt.%, which suggested that the corrosion resistance was improved. 

The equivalent circuit model A (Figure 7a) was used to simulate the EIS data at the initial stage of corrosion of the 15 wt.% and 20 wt.% AT13 composite coatings, which is composed of the resistance of electrolyte solution (R_s_), the pore resistance (R_p_) produced by the formation of the ionic conduction path through the coating and the constant phase element (CPE-c). Model B (Figure 7b) was applied to the EIS data simulation of the Fe-based coating, the 5 wt.% and 10 wt.% AT13 composite coatings and later-stage corrosion of the 15 wt.% and 20 wt.% AT13 composite coatings. For model B, CPE-ct represents a double-layer capacitance between the oxide layer/coating interface. R_ct_ represents charge transfer resistance. Their expressions are as in Equation (1): (1)$$Z=1/Y_{0}{\left(j\omega \right)}^{n}$$
where *Y*_0_ and *n* are the two parameters of *Z*, respectively, *Y*_0_ represents the admittance constant and *n* is the dispersion index of *Z*. The value of *n* is set between 0 and 1.

The variation of electrochemical parameters for the EIS spectra of the coatings with corrosion time is shown in Figure 8. At the initial stage, the maximum value of R_p_ was 8.51 × 10^3^ Ω·cm^2^ (Figure 8a), which implied that dense passivation was produced in the coating surface to prevent Cl^−^ ion erosion. Compared to other coatings, the corrosion resistance of the 15 wt.% AT13 composite coating was the best. As the immersion time increased, the value of R_p_ decreased sharply and then remained steady, indicating that the corrosion products dissolved into the solution gradually [14]. Figure 8b shows that the values of CPE-c increased significantly when it increased to 21 days, suggesting that the protective ability of Fe-based and 5 wt.% AT13 composite coatings greatly weakened due to the increase of coating porosity. In Figure 8c, we can see that the values of R_ct_ increased correspondingly with the addition of ceramic phase, which indicated that the composite coating enhanced a hindrance to the corrosion solution. This proved that the corrosion resistance of the 15 wt.% AT13 composite coating is more superior to the others, which is consistent with the polarization result.

### 4.2. Analysis of the Surface Element

In order to further analyze the corrosion mechanism of different coatings, the Fe-based amorphous coating and 15 wt.% AT13 composite coating were investigated by XPS. Figure 9 presents the XPS fine spectrum of Fe2p, Cr2p and Mo3d in the surface passivation. As shown in Figure 9a,b, the Fe2p_3/2_ spectrum consists of four overlapped peaks at 707 and 709~714 eV, which matches the Fe^2+^ and Fe^3+^ oxidation states [13]. In Figure 9c,d, the Cr2p_3/2_ spectrum can be resolved into three peaks at 574, 576 and 578 eV. The peaks at 574, 576 and 578 eV represent the Cr^0^ metallic state, Cr^3+^ oxidation state and Cr^6+^ oxidation state, respectively [15]. Moreover, the Co^0^ metallic state is a part of the Cr2p_3/2_ spectrum. The fitting results of the Mo3d spectrum are shown in Figure 9e,f. Meanwhile, two peaks at 228~229 and 230~231 eV were attributed to the Mo^0^ metallic state. The peaks at 229~230 and 232~233 eV could be associated with the Mo^3+^ oxidation state. The peaks at 233~234 and 234~236 eV were attributed to the Mo^4+^ oxidation state.

The quantitative analysis of XPS results can help to easily understand the corrosion resistance mechanism of the coatings. We suppose that the metal ions in the oxidation state are derived from the composition of the passivation film, and the metal ions correspond to the passivation film/coating metal interface. The relative composition of passivation film and the interface can be determined. Figure 10 shows the cationic component in the passivation film and the atomic component at the coating interface. For comparison, the passivation film of the pure coating exhibited a higher Fe oxide content and a lower Mo oxide content than the composite coating. In addition, the Mo3d spectrum of the composite coating surface was mainly the Mo^4+^ oxide state. It was reported [16] that Mo^4+^ oxide in the passivation film can greatly improve the stability of the passivation film, thereby further enhancing the local corrosion resistance. In this study, the corrosion resistance of the composite coating was better than that of the single coating. This result is consistent with the EIS data mentioned above.

### 4.3. Analysis of the Corrosion Morphology

To clarify the mechanism of corrosion for the coatings, the surface corrosion morphologies of the pure Fe-based coating and that of the 15 wt.% AT13 composite coating after immersion for 28 days were investigated by SEM. As shown in Figure 11, the corrosion of the coating surface was uneven. In addition, the corrosion preferentially occurred at the edge of the pore defect and the oxide [17]. Nevertheless, the corrosion of the composite coating was less serious. This is because many pores and micro-cracks occurred in the single coating, and the Cr-rich and Cr-poor regions at the oxide edge formed the micro-galvanic corrosion, which will become a preferential corrosion region, leading to the ion penetration and corrosion reaction. As the corrosion time extended, the coating peeled off and formed obvious pits. This indicates that the corrosion resistance of the coating was further deteriorated, but a large amount of corrosion products have not yet accumulated.

## 5. The Corrosion Mechanism in NaCl Solution

The long-term corrosion mechanism diagram of the composite coating is illustrated in Figure 12. As shown in Figure 12a, there are some pore defects on the surface of the sample. Meanwhile, there are mainly some interlayer gaps composed of oxide. Due to the flying oxidation of particles during plasma spraying, Cr-depleted areas are formed around the pores [18]. However, when the coating was fully immersed, the pores on the surface were filled with NaCl solution. After that, the Cr-depleted area around the pores quickly dissolved (shown in Figure 12b), thus showing a typical crevice corrosion reaction at the micro-defects of the pores. At this time, the surrounding area will be further dissolved because of the location advantage of the extreme occlusion in the hole and the harsh corrosion environment. As the immersion time increased, the corrosion area of the pores increased and the depth of the corrosion also increased. It gradually connected with the internal closed pores and the intermediate layer, becoming a channel for corrosion ions to invade the coating/substance interface (Figure 12c). In general, with the increase of AT13 content, the pore micro-defects and pore size in the composite coating reduced. It resulted that the probability and rate of the crevice corrosion reduced, and the polarization test also showed lower corrosion current density and higher self-corrosion potential.

## 6. Conclusions

The composite coatings with various proportions of AT13 ceramic additions were fabricated by thermal spraying technology. The corrosion resistance of the coatings in 3.5 wt.% NaCl solution was investigated and the corrosion mechanism was proposed according to immersion tests. The results are summarized as follows.

As the content of AT13 increased to 15 wt.%, the corrosion current density of the coating reached the lowest and the self-corrosion potential was the highest, showing the obvious passivation behavior in 3.5 wt.% NaCl solution. It indicates that the addition of AT13 phase effectively improved the corrosion resistance of the coating in simulated seawater.

Long-term immersion corrosion was carried out on different coatings. The equivalent circuit of the pure coating and the 5 wt.% and 10 wt.% AT13 amorphous coatings was R(Q(R(QR))), and the equivalent circuit of the 15 wt.% and 20 wt.% AT13 amorphous coatings increased (R(QR)) to (R(Q(R(QR))). This showed that the protective effect of the coating was enhanced with the increase of the amount of AT13.

## Figures and Tables

**Figure 1 micromachines-13-00056-f001:**
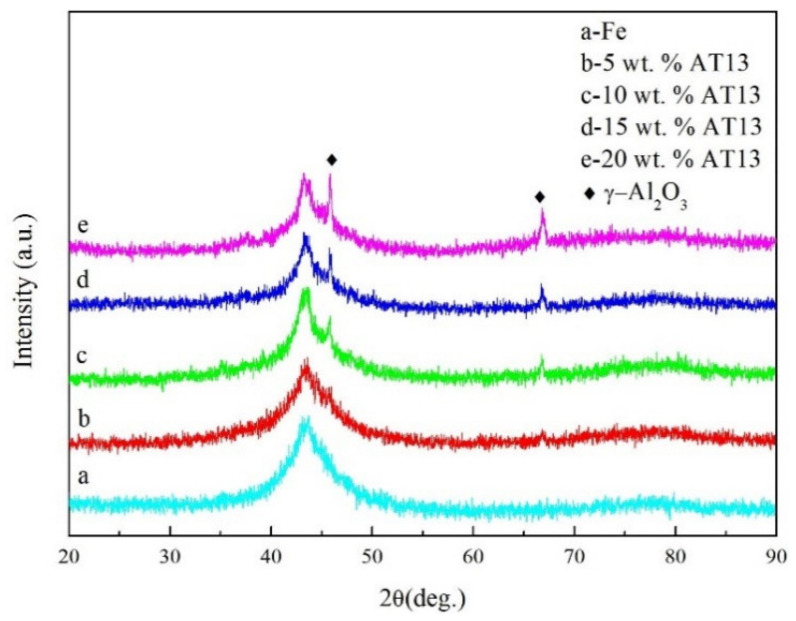
The XRD patterns of composite coatings.

**Figure 2 micromachines-13-00056-f002:**
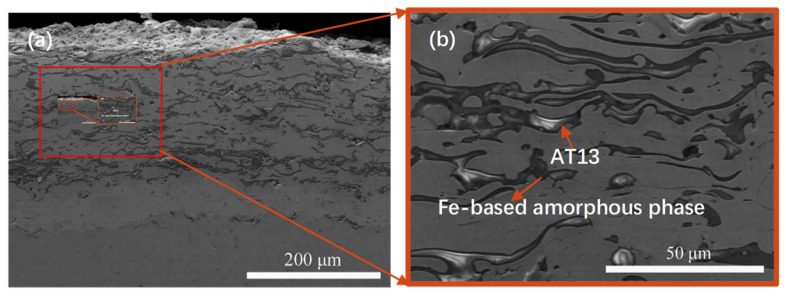
SEM images of (**a**) the cross-section of the composite coating with 10 wt.% AT13, and (**b**) the magnified image of the region marked in (**a**).

**Figure 3 micromachines-13-00056-f003:**
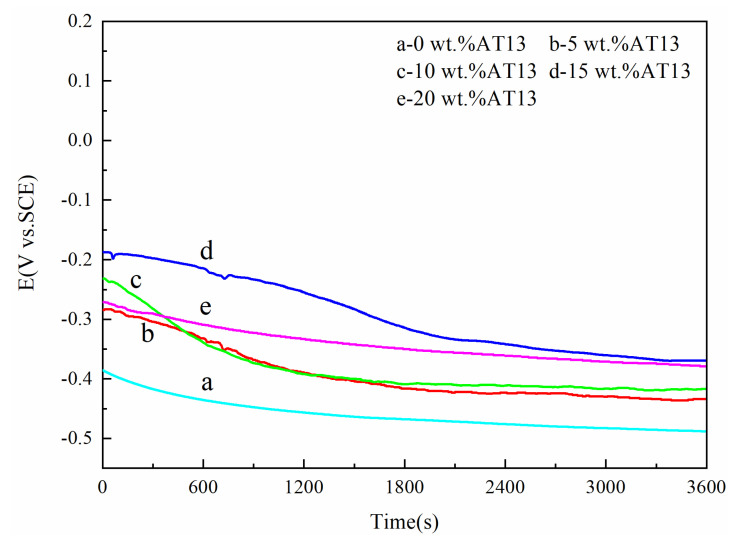
The open circuit potential of different samples and times in 3.5 wt.% NaCl corrosive solution.

**Figure 4 micromachines-13-00056-f004:**
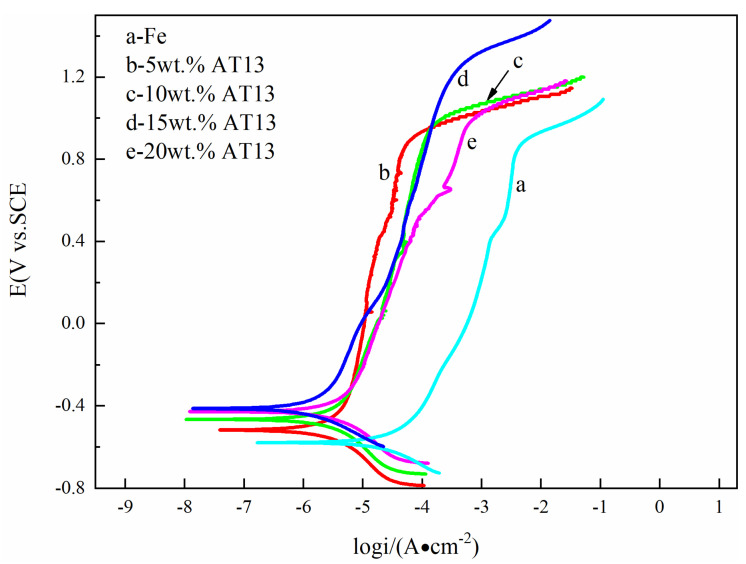
Potentiodynamic polarization curves of different samples in 3.5 wt.% NaCl solution.

**Figure 5 micromachines-13-00056-f005:**
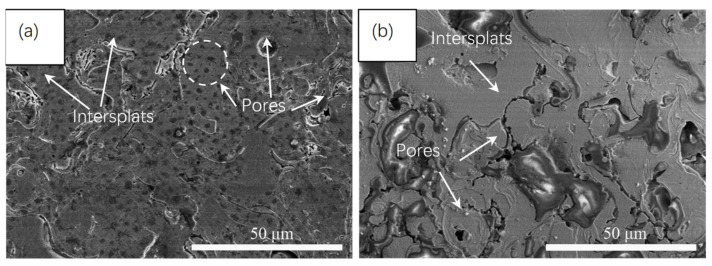
SEM images of potentiodynamic polarization of different samples in 3.5 wt.% NaCl solution: (**a**) Fe-based amorphous coating, and (**b**) 15 wt.% AT13 composite coating.

**Figure 6 micromachines-13-00056-f006:**
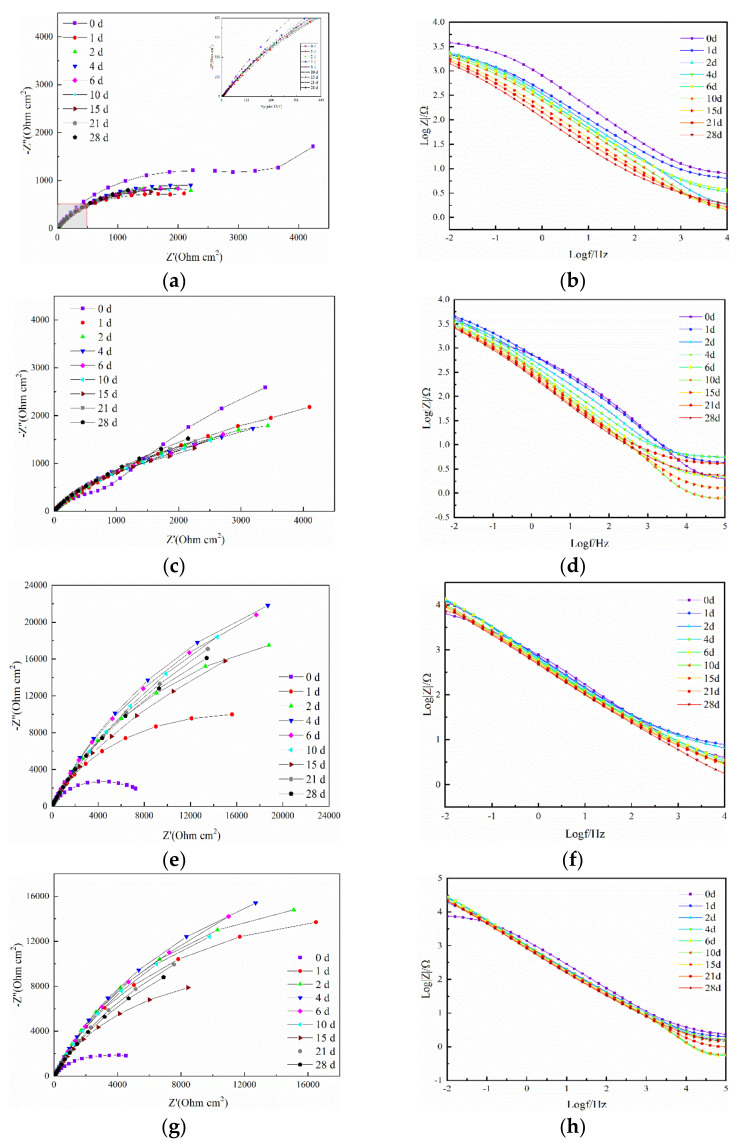
EIS plots of pure Fe-based coating, 5 wt.%, 10 wt.%, 15 wt.% and 20 wt.% AT13 composite coatings after different immersion times: (**a**,**c**,**e**,**g**,**i**) Nyquist plots, and (**b**,**d**,**f**,**h**,**j**) Bode plots.

**Figure 7 micromachines-13-00056-f007:**
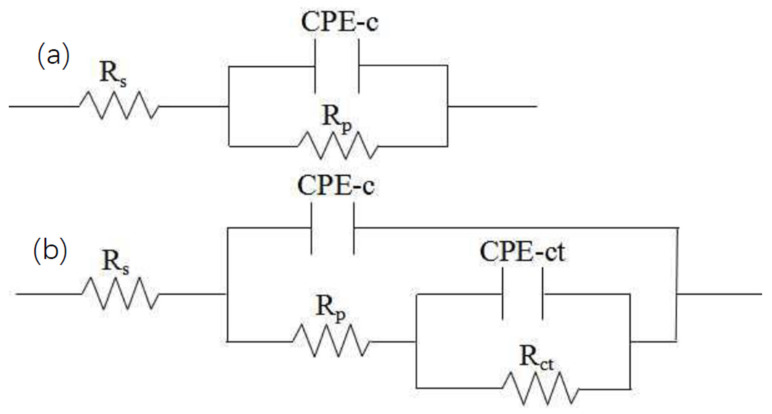
The fitted equivalent circuits: (**a**) initial stage corrosion of 15 wt.% and 20 wt.% AT13 composite coatings, and (**b**) Fe-based coating, 5 wt.%, 10 wt.% AT13 composite coatings and later stage corrosion of 15 wt.% and 20 wt.% AT13 composite coatings.

**Figure 8 micromachines-13-00056-f008:**
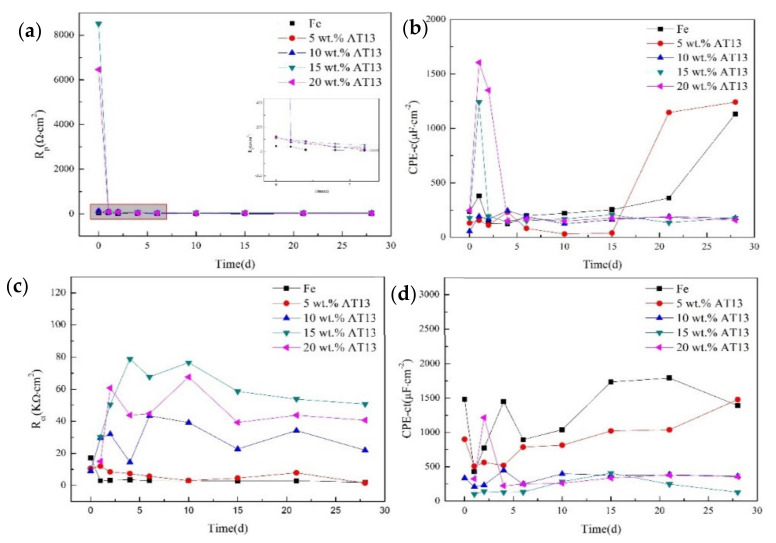
The variation of electrochemical parameters for the EIS spectra of coatings with corrosion time: (**a**) R_p_ time curve, (**b**) CPE-c time curve, (**c**) R_ct_ time curve and (**d**) CPE-ct time curve.

**Figure 9 micromachines-13-00056-f009:**
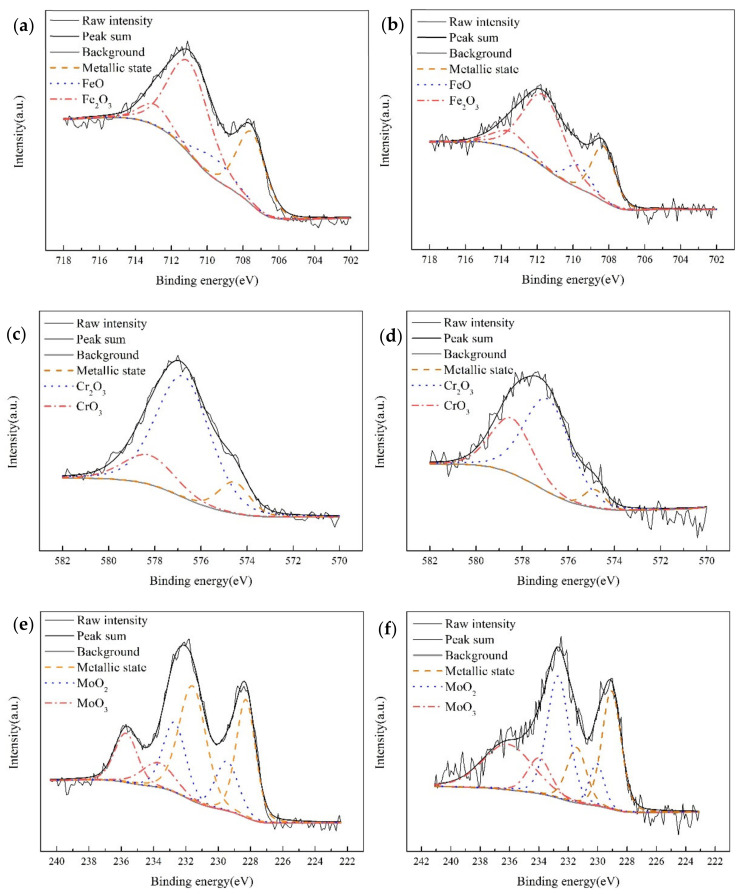
XPS spectrum: (**a**,**c**,**e**) amorphous coating, and (**b**,**d**,**f**), 15 wt.% AT13 composite coating.

**Figure 10 micromachines-13-00056-f010:**
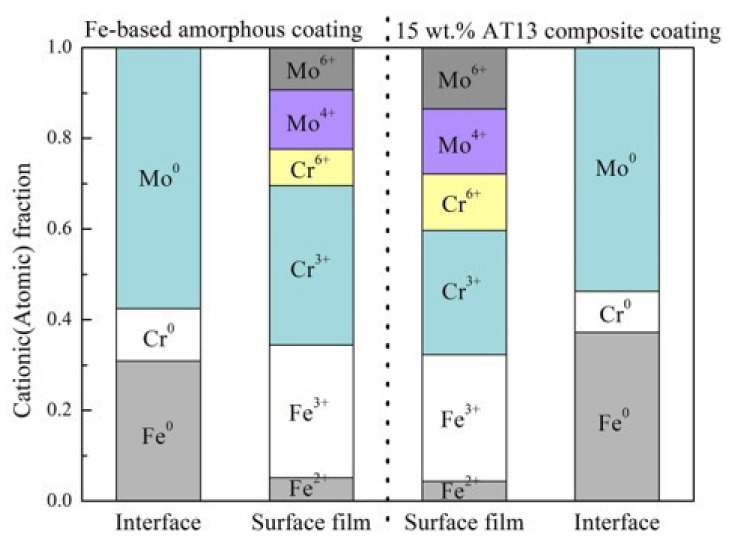
Cationic fractions of the oxide species and atomic fractions of the metallic species for the detected Fe, Cr and Mo elements by XPS from different coatings.

**Figure 11 micromachines-13-00056-f011:**
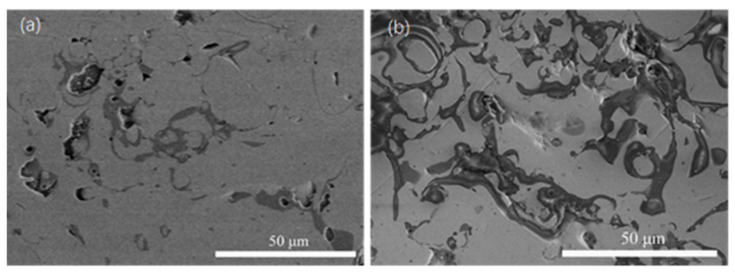
Corrosion morphology after immersion in 3.5 wt.% NaCl solution for 28 days: (**a**) pure Fe-based coating and (**b**) 15 wt.% AT13 composite coating.

**Figure 12 micromachines-13-00056-f012:**
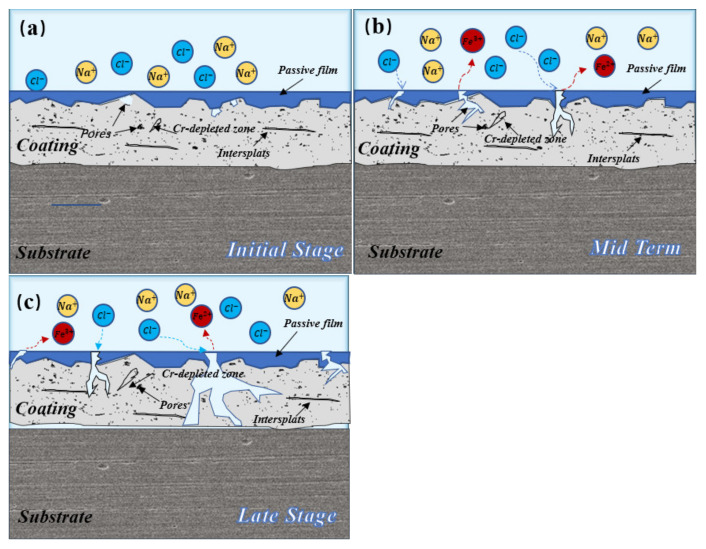
The long-term corrosion mechanism diagram of the composite coating: (**a**) initial stages of corrosion, (**b**) middle stages of corrosion and (**c**) late stages of corrosion.

**Table 1 micromachines-13-00056-t001:** Summary of the electrochemical parameters.

Corrosive Medium	Coating	E_corr_/mV	i_corr_/A cm^−2^	i_pass_/A cm^−2^	E_tr_/mV
3.5% NaCl	Fe	−580 ± 0.1	5.14 ± 0.30 × 10^−5^	5.4 ± 0.10 × 10^−3^	867 ± 0.08
	5 wt.% AT13	−519 ± 0.08	7.50 ± 0.25 × 10^−6^	6.23 ± 0.08 × 10^−4^	905 ± 0.10
	10 wt.% AT13	−467 ± 0.05	5.01 ± 0.28 × 10^−6^	5.12 ± 0.12 × 10^−4^	993 ± 0.09
	15 wt.% AT13	−411 ± 0.02	1.75 ± 0.12 × 10^−6^	5.37 ± 0.06 × 10^−4^	1236 ± 0.02
	20 wt.% AT13	−430 ± 0.03	4.06 ± 0.18 × 10^−6^	6.94 ± 0.15 × 10^−4^	1012 ± 0.08

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
