# Peer review of "The Corrosion Resistance and Mechanism of AT13/Fe-Based Amorphous Composite Coatings"

_micromachines, 2021, doi:10.3390/mi13010056_

Round 1

Reviewer 1 Report

In the artikle present study, the corrosion resistance of amorphous coating and composite coatings in 3.5 wt.% NaCl, 0.5 M H2SO4 and 10 wt.% NaOH solution were studied.
The corrosion resistance of amorphous and composite coatings with various methods is exhaustive and professionally carried out. Demonstration that when the content of the AT13 additive (Al2O3–13% by weight of TiO2) increases, the corrosion resistance increases (e.g. at 15% by weight, the composite coating had the lowest corrosion current density (1.75 × 10-6 A / cm2) ), is a very interesting and useful result both in the scientific and industrial aspect.

Reviewer 2 Report

Comments:

- The electrochemical test is not the Tafel extrapolation, but the potentiodynamic polarization. In the Tafel extrapolation, the polarization range is 300 mV below and above the open circuit potential. 

  • How did the authors measure the porosity of coatings?
  • The ipass is not the transpassive current density but the passive current density obtained at the middle of the passive region. 
  • If the ipass of Fe in the neutral and saline medium and in the acid medium is 10-3 A/cm2, did the authors consider that passivation occurred? I think in this case, the iron did not passivate. 
  • Fig. 3 - If the transpassive potential is 1.0 V or more (1.2V) , I think that the current density increases due to the oxygen evolution. Please, review and discuss. 
  • Page 4, line 128: This ipass is of iron, it is not of the coating. Please, review. 
  • Fig. 7 (a,c) : The Nyquist diagram must be square with the same scale in x and y axes. 
  • Page 9, lines 217 and 218: The impedance values were reduced after what??? Please, explain. 
  • Page 9, line 236: This diagram is not a Bode plot but a Nyquist diagram. Please, correct. 
  • Please, explain: If the sample did not show the Warburg impedance and diffusion control of corrosion, it signifies that the sample showed a high corrosion resistance???? 
  • Fig. 11: Please, correct all superscripts in Fig. 11. There are Cr 3-, etc
  • 8 references among 19 references are from 2016 to 2021. Please, cite new references. 

Reviewer 3 Report

Reviewer Comment for Author:

  1. Qualitative information’s are missing in abstract. Abstract should be to improve with more specific results.
  2. All acronyms should be introduced at first place of appearance in the text. For example WC (line 35).
  3. The work in a basic environment shows that the metal becomes passive according to the Pourbaix diagram. Justifies the use of NaOH
  4. To compare its two images it is necessary that it is of the same size (100 mm) in more, it is necessary to add the spectrum EDX. You say that the porosity is 4.3 % how calculate this value
  5. Concerning the polarization curves, you only draw the cathode branches. To see what happens in the anode domains or less arrives up to 1200 mV. Calculate the inhibition efficiency
  6. In potentiodynamic polarization studies, variation of open circuit potential with time should be added.
  7. In Tables 2 check the amounts of errors (±).
  8. The axes of figures 7 are not orthonormal. The frequencies and the Fitting curve in Nyquist plots should be included.
  9. The electrical circuits you used are not correct, so they should be corrected
  10. Recorded the parameters of the 8 curves in a table and added the equations you used to calculate the associated capacities and check the error quantities (±)
  11. The conclusion is very poor, it needs to be improved
  12. Compare your results with literature ones, and add recent references with its doi

Round 2

Reviewer 2 Report

Comments:

  • The authors said in the text that the silver phase in Fig. 2 (b) is alumina but in the figure, it is ZrO. Please, correct.
  • - Page 3, line 107: How did the authors measure the porosity? In methodology, this procedure is missing. 
  • Page 3, line 122: The authors did not say in the methodology that they studied 316L steel. Please, complete the experimental procedure. 
  • In Table 1, the authors should present the passive current density that is an important parameter because the materials showed passivation. 
  • Considering the breakdown potential, composite coated steels showed a higher corrosion resistance than the 316 steel. but considering the passive current density, the 316 steel showed a better corrosion behavior than the composite coated steels. Please, discuss. 
  • Please, in Fig. 4 what is the potential of the oxygen evolution? At higher potentials, the current can increase due to the oxygen evolution and not necessarily due to the degradation of the passive layer. Please, review. 
  • Fig. 5: Did the authors observe pits on the surface of Fe amorphous and composite coated steels after the polarization tests?
  • The procedure of immersion tests is missing. Please, complete the information in the methodology. 
  • Page 6, line 218: The diagram is the Nyquist and not the Bode. Please, correct. 
  • Page 6, line 227: The diagrams are Nyquist and not Bode. Please, correct. 
  • Page 6, line 237: The presence of the Warburg element only indicates the corrosion mechanism, controlled by diffusion, and not if the corrosion resistance is higher or lower. Please, review. 
  • Page 8, line 269: The circuit is shown in Fig. 7 (a) only contains Rs, CPE-c and Rp. Please, correct. 
  • 11 references among 19 are from 2016 to 2021. 

Reviewer 3 Report

Reviewer Comment for Author

The Nyquist curves and the electrical circuit are not compatible, there is something wrong, so:
puts the frequencies in the Nyquist curves and checks the calculations of the capacities of the two loops
the existence of two loops in the Bode representation  is quite clear but in the Nyquist is not the same
Polarisation resistance is the sum of all resistances and resistances are differentiated by its associated capacity values. in this case, they are the same which shows that we have two Rct
check your calculation, you probably find Rf and Rct (Cf less than 100 the Nyquist curves and the electrical circuit are not compatible, there is something wrong, so:
puts the frequencies in the Nyquist curves and checks the calculations of the capacities of the two loops
the existence of two loops in the Bode representation  is quite clear but in the Nyquist is not the same
Polarisation resistance is the sum of all resistances and resistances are differentiated by its associated capacity values. in this case, they are the same which shows that we have two Rct
check your calculation, you probably find Rf and Rct (Cf less than 100 µF))

Round 3

Reviewer 2 Report

Comments:

-Fig. 2 (b) The authors indicated ZrO but it is Al2O3!!!! Please, correct.

  • Page 4, line 144: Please, correct: passive current density, passive potential (Epass)....
  • - Page 4, line 159: The 316L steel showed the lowest corrosion current density but mainly the lowest passive current density indicating a more protective passive layer. Please, discuss. 
  • - Fig. 4: I don't understand: The Fe-based amorphous coating did not show uniform corrosion, it seems that a less protective passive layer was formed with a breakdown potential. The passive layer of Fe-based amorphous coating shows the highest passive current density and is less protective than the others. 
  • - In Table 1, the authors should give the passive current density that it is more important in this case than the corrosion current density. 
  • - Page 6, line 226: When the sample is polarized, it acts as an anode and the microcell or galvanic cell does not occur. 
  • - Page 9, line 319: I think which is consistent with the polarization result. Please, correct. 
  • The passive current density should be considered in the corrosion result discussion. As the breakdown potential is high, the authors should mention the potential of oxygen evolution ( water oxidation). 
